

# A mechanistic classification of double tides
J. A. Mattias Green[1], David G. Bowers[1], and Hannah A.M. Byrne[2]
[1]School of Ocean Sciences, Bangor University, Menai Bridge, UK
[2]Department of Evolutionary Biology, Uppsala University, Uppsala, Sweden.
Correspondence to: Dr Mattias Green (m.green@bangor.ac.uk)
**Abstract**
Double high or low tides are usually explained by adding a higher harmonic to the dominating tide. In its simplest form, the
criterion for a double tide is that the amplitude ratio between the higher harmonic and the dominating constituent is larger than
$1/n^2$ where $n$ is the ratio of their periods. However, it is not always clear how the higher harmonic becomes large enough to
generate the double tide. This is rectified here by identifying three possible ways to enhance the higher harmonic enough to
produce a double tide. Using TPXO9, the latest version of the altimetry constrained global tide database, potential locations
for all three classes are identified and the existence of double tides are then evaluated using historic long-term tide gauge data
from nearby locations. Thirteen locations with double tides were identified this way across the classes, of which seven are
discussed further and shown to fit the classification scheme. The search criterion for classes 1 and 2, based on the amplitudes
of M2, S2, and M4, work well with TPXO9 and suggests over 400 locations with double tides. The main reason we cannot
identify more double tide locations is a lack of TG data, especially in the polar areas. Class 3, which requires an embayment
resonant for the higher harmonic initially provided over 8000 potential locations, but only a few of these were in embayments.
This class thus requires more manual work to identify the locations. It is concluded that the mechanism behind double tides in
most textbooks needs to be revised because they are far more frequent in both space and time than previously thought.

**Introduction**
Double tides (DT in the following) are a quite rare but interesting feature of coastal tidal observations. In a double high the
tide rises to a first maximum, followed by a short dip in water level, before it rises again to a second maximum and then falls
towards the subsequent low water (Bowers et al., 2013; Byrne et al., 2017; Godin, 1993), whereas in a double low tide the ebb
tide starts to rise, only to drop back down before the rise starts anew. Double tides are usually not generated by the lunar or
solar astronomic forcing but rather by the addition of locally generated higher harmonics to the dominating tide. Furthermore,
DTs can only be generated if the phasing between the higher harmonic and the lower frequency tide (henceforth called the
parent constituent) is correct and the higher harmonic has a large enough amplitude (Byrne et al., 2017). We therefore expect
to find DTs only in shallow water, where higher harmonics can be generated due to frictional effects and the reduced depth
(Godin, 1993), or by a transient seiche set up by the tide (Bowers et al., 2013). The DT can be generated this way because the
wavelength, L, of a higher harmonics of the tide is shorter than that of a constituent with a lower frequency (i.e., $L_{M4} < L_{M2} < L_{K1}$).
This means that any location with a double high tide has a nearby location with a double low tide as well. Arguably, the best-
known double tide occurs in Southampton (UK), where the prolonged high water associated with the double low tide provided
the port with a commercial advantage. An example of a location with a double low tide is Providence in Narragansett Bay,
USA (Redfield, 1980). DTs have also been described in Den Helder in the Netherlands, Woods Hole, USA, Port Ellen, UK



(see Byrne et al., 2017, for the analysis of the latter), and in Daya Bay, China, where M6 is the responsible constituent (Song
et al., 2016).
Doodson proposed a minimum criterion to predict the occurrence of DTs tides when a higher-frequency tidal harmonic
(normally M4) is added to the semi-diurnal tide (M2):

$$\frac{b}{a} > \frac{1}{n^2} \tag{1}$$

Here, $b$ ($a$) is the amplitude of the higher (parent) harmonic, and n is the frequency ratio between the two (i.e., for M2 and M4,
$n = 2$). Byrne et al. (2017) expanded this criterion to include the phase difference between the two harmonics. The revised
parameter is $B = br^2/a$, where $r$ is a number generally less than $n$. They applied the new theory to data from Port Ellen in
Scotland and found that For Port Ellen, $B > 1$ and $n = 3$ (i.e., M6 is the source of the DT) when DTs occur. Note that throughout
most of the ocean, the ratio of amplitudes $b/a$ is small – typically less than 1% - and special conditions are needed to increase
the ratio above the critical value.
A fundamental question is how are the right conditions for double tides created? What is needed is to increase the size of the
higher harmonics or reduce the size of the lower, or both. Based on the recent investigations into DTs, we now have ability to
propose classifications for DTs based on different forming mechanisms. We have identified three possible mechanisms:
1.  The DT will occur close to a node in M2 and an antinode in M4, giving M4 the possibility to set up a DT by itself. This is
referred to a class 1 DT in the following, and it is the textbook mechanism behind the DT in Southampton.
2.  In the case of Port Ellen, the semi-diurnal tide is reduced at neaps because M2 and S2 are about the same size (Byrne et
al., 2017). If M4 is large, it can flatten out the ebb tide during neaps, giving higher harmonics the opportunity to set up a
double low tide, or M4 can do it by itself. We call this a class 2 DT in the following.
3.  In the third class, M4 is amplified more than M2 inside an embayment and cant therefore generate a DT. This can happen
if the bay is near-resonant for the higher harmonic. An example of a class 3 tide occurs in Providence, at the head of
Narragansett Bay (Redfield, 1980), and is analysed further below.
Double tides are rare but more are being discovered, e.g., Port Ellen and Daya Bay. The aim of this paper is therefore to carry
out a global search for the conditions likely to create a double tide. The criteria we use for this search are those that we would
expect for the three classes of double tide that we have described (see below for details). We use the latest altimetry constrained
tidal elevation database (TPXO9) in combination with historic tide-gauge data to locate and classify locations with double
tides. Potential locations for class 1 and 2 DTs can be directly identified in the altimetry data (see below for details). Class 3
double tides are more difficult to locate in this way because they depend on the characteristics (size and depth) of a coastal
embayment in which the port is located. Suggested locations can still be found in the altimetry data, but most will have to be
discarded and we will therefore deal with class 3 double tides on a case-by-case basis.
**Data and analysis**
To pinpoint possible locations of DTs we use TPXO9 (see http://volkov.oce.orst.edu/tides/global.html and Egbert and
Erofeeva, 2002) to map where we may expect DTs based on the classifications above. TPXO9 is the latest version in the
TPXO-series, which consists of global tidal solutions that fit, in a least-squares sense, both Laplace's tidal equations and
altimetry data. In the version used here, the data comes at a horizontal resolution of 1/6° in both latitude and longitude. The
criteria used for each class in the search were
Class 1: The ratio between the M4 and M2 amplitudes are larger than 1 and M2 is smaller than 0.025m.
Class 2: The ratio between M2 and S2 is larger than 0.9, and M2 is larger than 0.1m and M4 is larger than 0.025m,



Class 3: The ration between M4 and M2 is larger than 0.1 at the mouth to a bay or gulf. Note that from the TPXO data we have a lot of points where this is fulfilled, but along coastlines and this class require substantial manual identification of locations.

For all classes we also required that the local water depth, at the TPXO9 resolution of 1/6°, was shallower than 500m. This made sure that amphidromic points in the open ocean were excluded.

We then used the GESLA tide gauge data set (see Woodworth et al., 2017 and http://gesla.org for details) to compute the amplitudes and phases of the K1, O1, M2, S2, M4, M6, and M8 constituents for the TG station nearest to the potential DT-locations identified in the TPXO-data. Note that in some areas, e.g., in Antarctica, along the Siberian coast, and in Hudson Bay, there are potential DTs, but no suitable TG stations nearby. The TG data was analysed using t_tide (Pawlowicz et al., 2002). In the case of multi-year data sets, the harmonic analysis (HA) was done on a selected year, preferably towards the end of the series, where DT were found through visual inspection. For clarity, we only show data from a single 2-day period, during which DTs were found, in the following. Further analysis of specific periods took place for class 3 DTs and are described below.

## Results

### TPXO results

We identified 219 class 1 locations (see Figure 1a), 140 potential class 2 locations (Figure 1b), and over 8000 class 3 points (Figure 1c). The latter could quickly be whittled down to about 100 locations because the initial criteria stipulates that we should be in an embayment – something we didn't take into account when doing the initial search in Figure 1c. The locations of potential DTs where then reduced further by our requirement that there must be a GESLA TG within 20km (corresponding roughly to the width of one TPXO9 gridcell) from the TPXO9 DT point. This led us to finally process data from 30 TG, and we found double tides at 13 locations, including the 4 previously reported on. From these, we opted to further discuss seven: Den Helder, Providence (and Newport, although there is no DT there – see the section on class 3 DTs for details), Thevenard, Victor harbour, Rio Grande, and Imituba. The results of the HA is summarised in Table 2, and the locations of these gauges are marked in Figure 1b

### Class 1

Class 1 DTs, i.e., when the higher harmonic is larger than the parent because of an amphidromie of the parent is located nearby, were found at two new locations: Pari in Indonesia and Rio Grande in Brazil, along with the previously reported double high tide in Den Helder in the Netherlands (see Figure 2). Rio Grande is a special case, where the double tide, and on occasion triple highs, occur on the dominating diurnal tide. Because of this uniqueness we discuss it separately later. The DT at Pari is quite small, and instead we choose to discuss Den Helder.

This location experiences a double high tide because it is close to an M4 antinode and an M2 node. In fact, we are between to M2 amphidromes, but because of the nature of the tide in the North Sea, that corresponds to a tidal minimum (Figure 2a). This local minimum in M2 tidal amplitudes allows the relatively large M4 in the region to set up the double high tide seen in Figure 2c (see also Figure 2b).



**Class 2**

Class 2 DTs appeared in 7 locations, including the previously described Port Ellen. Note that Rio Grande may be either a class 1 or class 2, or both, and is discussed further below). At these locations there is a large spring-neap cycle and the overtides, sometime multiple constituents in partnership, can generate DTs. Two clear examples can be found on Australia's south coast, where double low tides are found at the data from Thevenard and Victor Harbour (Figure 3a and b, respectively).

The data from Thevenard shows a flattened high tide during neaps, which M4 is able to modulate into the double low tide (M6 and M8 are both negligible at this location). Victor Harbor is more complicated, because we find the double low tide on the diurnal tide (dominated by K1). The mechanism is the same, however: there is a diurnal spring-neap cycle, the semi-diurnal tide, strongly modulated by the semi-diurnal spring neap cycle because S2 is larger than M2, can set up double low tides on the dominating diurnal tide. Note that K1 and S2 alone cannot generate a double low tide here; combined they give a very large daily inequality in the tide – and having the spring-neap modulation is a necessity to generate a genuine double low tide.

**Class 3**

Narragansett Bay in New England USA provides an excellent example of a class 3 double low tide. The mechanism proposed for this class is that the M4 tidal amplitude is amplified more in the bay than M2. In Narragansett Bay this happens because the natural resonant period is ~4 hours (L~35 km, H~8m, giving quarter-wavelength resonance at 4.4 hours). Using GESLA data from Newport (at the mouth) and Providence (at the head; see Figure 5) from 2014, we find that there is indeed amplification at the head of the bay: b/a=0.18 compared to 0.11 at the mouth. However, this is not enough to produce a double low tide, yet one is seen in the record at numerous times. This is because the annual fits are not necessarily representative of the conditions when the double lows occur – not all low tides have a double low in the Bay.

Instead, we focus our attention on the neap period around year day 44.2 shown in Figure 5, when we have an observed double low. Redoing the fit on a 25-hour part of record, between days 43.50-44.535, we find that the reduced semi-diurnal constituent indeed does allow for a double low: b/a = 0.26 at Providence, whereas it is 0.14 at Newport for this period, i.e., an amplification of ~85%. There are two conclusions to draw from this analysis: first, the mechanism suggested – M4 amplification - is the true one in this instance, and second that using long-term records to locate DTs, may be dubious because the annual fits may not necessarily capture the needed signals. Rather, looking at the semidiurnal neap amplitude (i.e., M2-S2) may be a better measure in that case. Indeed, the annual fit gives $b/a$ = 0.23 at Providence if the M2-S2 amplitudes are used for a. This further stress that it is at neap tides M4 can act in Providence and set up a double low tide.

Of course, Narragansett Bay is not unique in amplifying M4 more than M2 - any coastal bay with a natural period closer to 6 hours than 12 will do this (e.g., Song et al., 2016). What is special about Narragansett Bay is that the $b/a$ ratio at the mouth is so large and the phase is often just right for producing a flattening at low water. An amplification of $b/a$ by 50% then turns the flattened low water at Newport into a double low water at Providence. There is, however, more to the story in Narragansett Bay. Of the 673 tidal cycles in 2014 that we have analysed, 196 fulfil the Doodson criterion of b/a>0.25, but only 43 exhibit a double low water at Providence that would satisfy a visual inspection. Of these, 23 are produced by M2 and M4 alone, i.e., by the $b/a$ amplification larger than 50% between Newport and Providence. In the remaining 20 cases of a double tide in this period, the $b/a$ amplification serves to increase the duration of low-water flattening at Providence and then M6 acts on the flattened low tide to create a double low water. An example of this is shown in Figure 5 at day 327.8. In this case, there is only a 22% $b/a$ amplification, and although the ratio is supercritical at Providence, the *phase* is not right between the M2 low tide





and the M4 high tide. Consequently, we instead see an extended period of flattening of the low tide, on which the small 5cm
M6 tide can set up a small double tide. This is why the Doodson criterion is not always correct, and the phasing should be
taken into account as well (Byrne et al., 2017).
Note that there is no M6 tide at Newport (not shown). The M6 constituent at Providence must therefore be created within the
Bay. This happens because the resonance period of the Bay is about 4 hours, so any small forcing with this period is likely to
produce an amplified response in the Bay. The mechanism is known as a compound tide. When a shallow bay is forced with
a mixture of semi-diurnal and quarter diurnal tides, shallow water effects create tides with a frequency equal to the sum and
the difference of the frequencies of the two constituents: the compound tides thus have frequencies of $2\omega-\omega=\omega$ (which enhances
the semi-diurnal tide) and $2\omega+\omega=3\omega$, which is the M6 tide in this case. The amplitude of this new M6 tide is expected,
theoretically, to be proportional to the product $ab$ of M2 and M4 tides. This seems to be true in Narragansett Bay (not shown;
the M6 amplitude in providence is ~0.3$ab$) and is likely the mechanism in Daya Bay, China (Song et al., 2016).
To summarise, there are two ways of creating a class 3 DT. The first is by selective amplification of the M4 tide to the point
where the sum M4+M2 at the head is alone sufficient to make a DT. Here, this happens when the $b/a$ ratio at the mouth
(Newport in Narragansett Bay) reaches its largest values, which are then amplified at the head.. This mechanism is seen at
small neap tides. Alternatively, a lower value of $b/a$ at the mouth can be amplified to produce a low tide at the head with an
extended period of flattening. A small high frequency (M6) oscillation added to this will then produce a double-dip low tide.
This oscillation is likely to have a period of 4 hours, since that is the resonant period of the bay, and can have a variety of
causes.
Because the DTs here are more prominent during neaps than springs, we also asked if the 18.6-year lunar nodal cycle influence
the occurrence of DTs at Providence. This could be the case here because S2 does not have a nodal cycle so one could expect
that there would be a further potential for DTs during the nodal minimum in M2 tides, because the tidal range at neaps would
then be smaller. The results, which are not shown, suggest that there is no difference in the number of DTs occurring. The
reason is quite simply that because M2 is reduced, so are the overtides M4 and M6 and there is no net gain in the potential to
generate DTs.

### Diurnal multiple high- and low tides

The data from Rio Grande shows pronounced double low tides, and there are triple high tides on occasion (Figure 6a). This is
an exciting location, because S2 is larger than M2 and O1 is the dominating signal (see Table 2 for a summary). The variance
capture of the harmonic analysis is quite low - about 20% of the variability with the 7 constituents – mainly because the TG is
near the mouth of the river, and freshwater fluxes are affecting the time-series. In fact, on at least one occasion the TG actually
dries out.
The large O1 and K1 constituents lead to a diurnal spring-neap cycle with a 13.4-day beat. The semi-diurnal tide, dominated
by S2 but with a large semi-diurnal spring-neap cycle as well, can then set up a double low tide in the diurnal tide ($b/a>0.25$
for both M2 and S2 over O1). However, M4 is actually larger than M2 at the location, so it can generate a double low tide on
the semi-diurnal tide. Combined, these double DT lead to a triple high/low in the diurnal tide, as seen between days 210-210.5
in Figure 7. This is not a one-off occurrence of a THT in the series; the triple occur on most high waters during diurnal neap
tides. Note that the reconstruction of the fit (the red line) has a damped down version of the THT in it, so the THT must come
from the constituents under investigation. The timing is crucial here: we need a diurnal neap in combination with a semi-



diurnal spring with M4 timed right. The beat between the diurnal spring-neap and the semi-diurnal spring-neap cycle is 6.1
hours, which is very close to the period of M4, so this may exaggerate the M4 signal and allow setting up the THT.
Interestingly, the TG at Imituba, further north along the Brazil coast, also have triple highs (Figure 6). There is, however, a
different mechanism at play here: M2 is now the dominating tidal constituent, but the station sees a large daily inequality
generated by O1 and a large spring-neap cycle due to a large S2 constituent. This allows M4 and M6 together to generate
double and triple highs on the semi-diurnal tide.
**Discussion**
Most textbooks on oceanography state that DTs are set up by a combination of M2 and M4, with a DT generated if Doodson's
criterion is fulfilled. There is, however, usually very little discussion about *how* M4 becomes large enough to fulfil the Doodson
criterion. Here, we suggest three potential mechanisms responsible for this amplification, and show that even higher harmonics
may be needed to get the desired results. We also argue that in the search for DTs, visual inspection of the TG data is crucial,
and that fits must be done on short periods of data to capture the correct dynamics; annual fits are not necessarily 100% relevant
for short periods of data. Furthermore, global altimetry products, e.g., TPXO9, are helpful to identify potential locations of
DTs, but at the end of the day, it is necessary to have high temporal resolution TG data to identify and quantify the DT.
Whilst we identified a large number of locations with potential DTs in TPXO9, we were limited in the areas we could analyse
because of limited TG coverage, especially in the Polar Regions, including Hudson Bay. However, we feel that we have had
good enough coverage to show that our classification works, and leave further analysis and searches for TG data for future
publications. Another future prospect is to investigate if DTs will become more frequent during future change due to sea-level
and warming. This has implications for mitigation purposes, because a prolonged high tide due to higher harmonics has the
potential to increase flood risk due to storm surges. From the most recent simulations of the effect of sea-level rise on tides
(Schindelegger et al., 2018) it appears that there are no changes in the number of potential locations fulfilling Doodson's
criterion, but the resolution of the model prevents any further detailed simulations. Instead, we argue that it be worthwhile to
do regional high-resolution simulations of the Patagonian Shelf and Gulf of Maine to see how the DTs there may be affected
in the future.

Acknowledgements: The GESLA data can be downloaded from gesla.org – see the data source table on the GESLA page for
details about contributors.





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

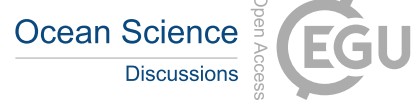



Tables

**Table 1: Results from the HA for the respective station. "Variance" refers to the variance captured by the harmonic analysis, using**
**O1, K1, M2, S2, M4, M6, and M8. "amp" refers tot the constituent's amplitude (in meters), whereas "pha" is the phase (in degrees),**
**measured relative to the starting point of the time series.**

|  | Den Helder Netherlands | | Providence USA | | Newport USA | | Thevenard Australia | | Victor Harbor Australia | | Rio Grande Brazil | | Imituba Brazil | |
|---|---|---|---|---|---|---|---|---|---|---|---|---|---|---|
| tide | amp | pha | amp | pha | amp | pha | amp | pha | amp | pha | amp | pha | amp | pha |
| O1 | 0.12 | 169.8 | 0.04 | 351.9 | 0.04 | 351.0 | 0.14 | 55.7 | 0.16 | 122.1 | 0.12 | 66.6 | 0.12 | 121.2 |
| K1 | 0.07 | 59.1 | 0.06 | 169.3 | 0.06 | 167.6 | 0.20 | 302.5 | 0.22 | 262.8 | 0.04 | 300.0 | 0.07 | 58.5 |
| M2 | 0.64 | 210.5 | 0.62 | 157.0 | 0.53 | 151.2 | 0.30 | 337.3 | 0.13 | 341.5 | 0.03 | 33.0 | 0.15 | 39.8 |
| S2 | 0.19 | 344.6 | 0.13 | 62.9 | 0.11 | 55.0 | 0.36 | 207.8 | 0.15 | 150.3 | 0.04 | 103.2 | 0.11 | 162.3 |
| M4 | 0.10 | 263.2 | 0.11 | 356.0 | 0.06 | 332.7 | 0.02 | 285.7 | 0.03 | 339.0 | 0.03 | 86.4 | 0.03 | 308.4 |
| M6 | 0.05 | 48.0 | 0.03 | 30.5 | 0.01 | 298.1 | 0.00 | 126.6 | 0.00 | 349.0 | 0.00 | 348.7 | 0.00 | 180.8 |
| Variance | 63.7% | | 85.8% | | 85.1% | | 70.9% | | 61.5% | | 19.3% | | 41.1% | |







**Table 2: Results from the HA on the data from Narragansett Bay. "Full year" refers to analysis of the whole 2014-record, "25-**
**hours" to the analysis of data from day 39-40.035, and "Annual Neap" to the results based on M2-S2 from the Full Year HA (see**
**the text for details).**

|  | Newport | Providence | Amplification |
|---|---|---|---|
| Full year |  |  |  |
| M2 | 0.53 | 0.62 | 1.17 |
| M4 | 0.06 | 0.11 | 1.83 |
| b/a | 0.11 | 0.18 | 1.64 |
| 25 hours |  |  |  |
| M2 | 0.36 | 0.42 | 1.16 |
| M4 | 0.06 | 0.11 | 1.83 |
| b/a | 0.17 | 0.26 | 1.52 |
| Annual neap |  |  |  |
| M2-S2 | 0.42 | 0.49 | 1.16 |
| M4 | 0.06 | 0.11 | 1.83 |
| b/a | 0.14 | 0.23 | 1.65 |






Figures

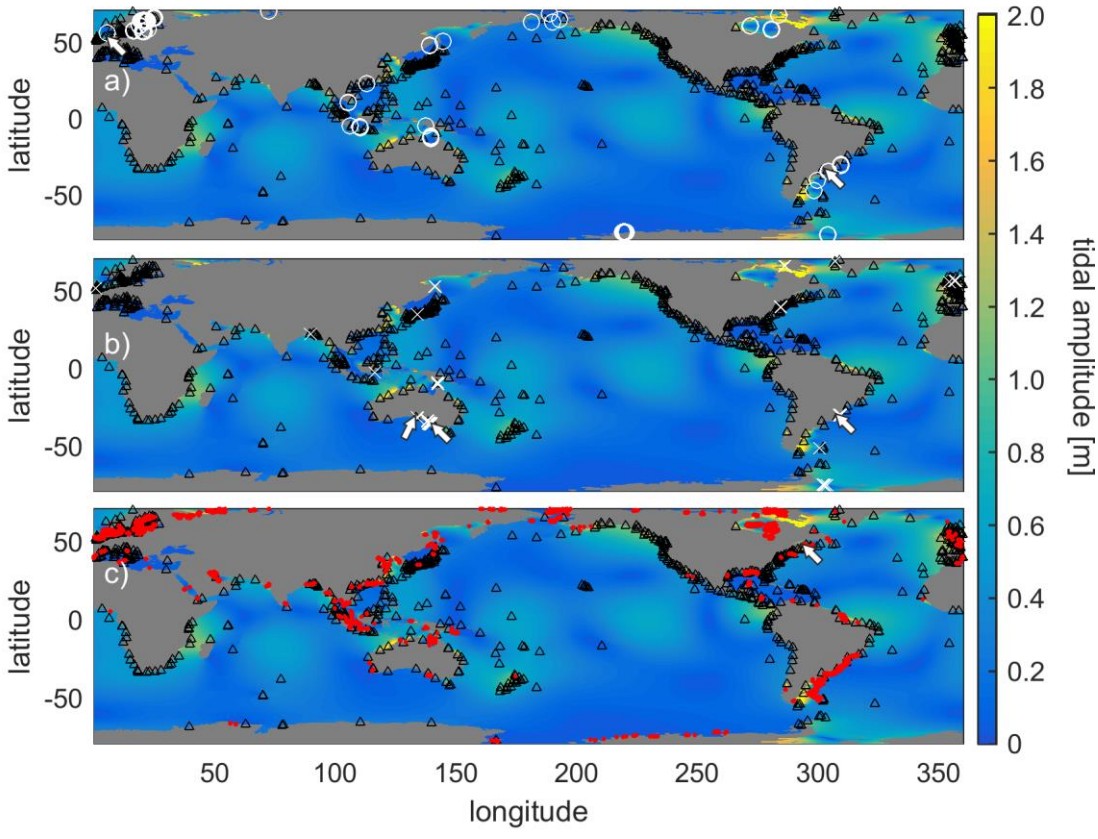


**Figure 1: M2 amplitudes (colour – note that the scale bar saturates at 2m) from TPXO and potential locations for DTs using the three classes (panel a)=class one, b)=class 2, c) = class 3). The TG locations in the GESLA database, in water shallower than 500m in the TPOX9 grid, are marked as black triangles. The filled white arrows point tow the location of stations analysed further in this paper.**

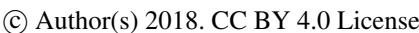



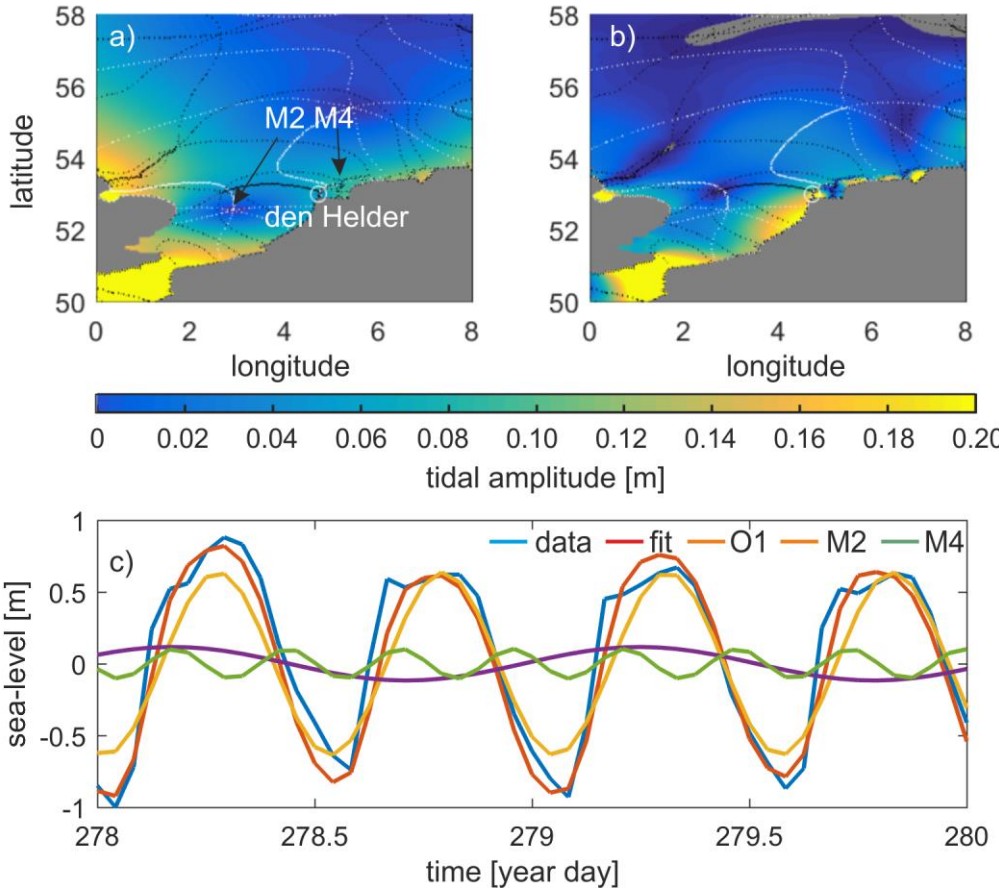


**Figure 2: a) A blow-up of Figure 1 showing the English Channel and the North Sea with Den Helder marked with a white circle. The colours show M2 (M4) amplitudes from TPXO9 in the left (right) panel, whereas the white (black) lines are M2 (M4) phases separated every 60°. The amphidromic points closest to the TG station are marked by the arrows.**

**b) As in a) but showing the M4 field.**

**c) Shown is an example of a class 1 DT in the TG data from Den Helder. The blue line shows hourly observations from 1986, and the red line the reconstruction after fitting O1, K1, M2, S2, M4, M6, and M8 to the data. The yellow, purple, and green lines are the fitted O1, M2 and M4 curves (see Table NN for a summary of the fit).**



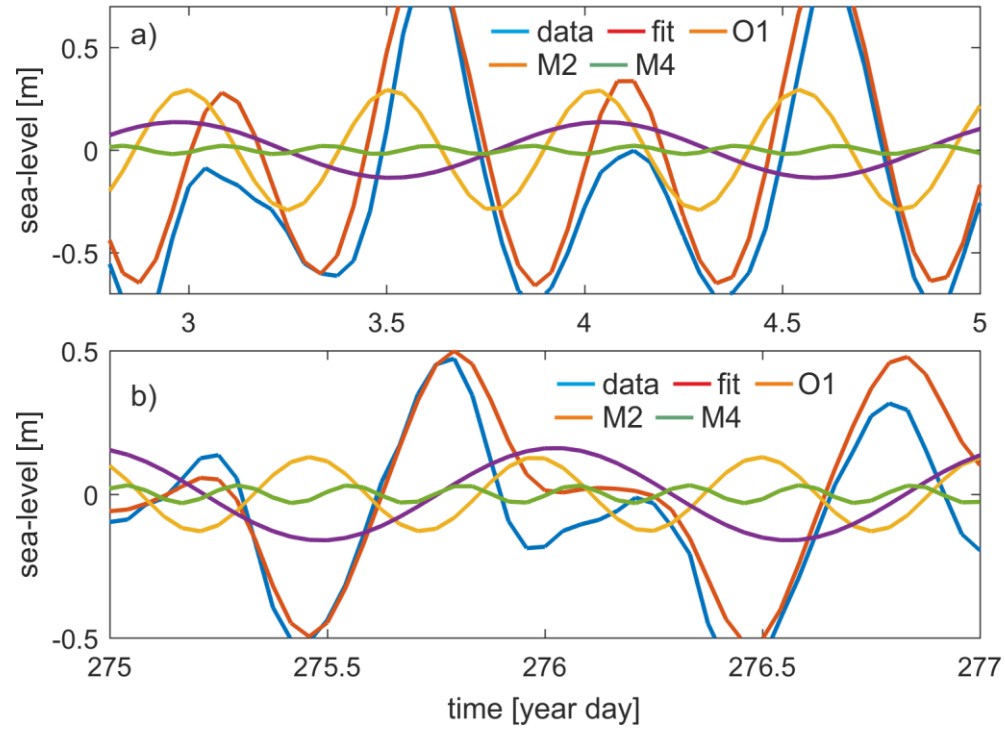



**Figure 3: As in Figure 2c but showing examples of Class 2 double tides from Thevenard (a) data from 2011) and Victor Harbor (b) data from 2004), both on the south coast of Australia.**





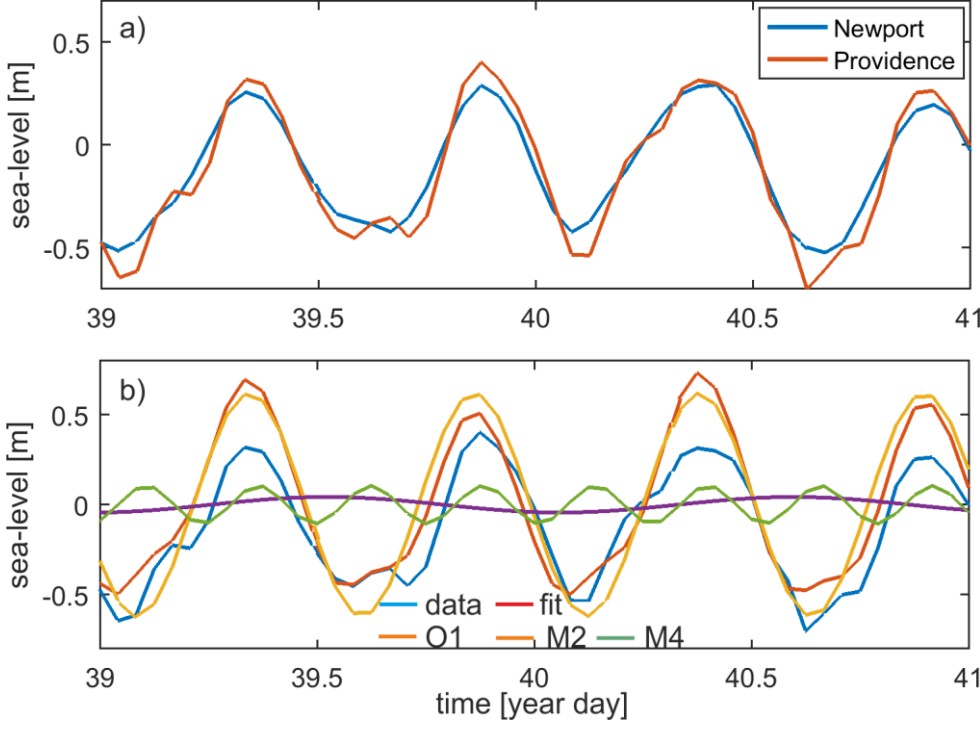


**Figure 4: A 2-day example of the sea-level record from Newport (blue) and Providence (red), USA, from 2014 (panel a). Panel b)**
**shows the Providence data and reconstructed results from the HA for the whole of 2014, i.e., as in Figure 2c.**






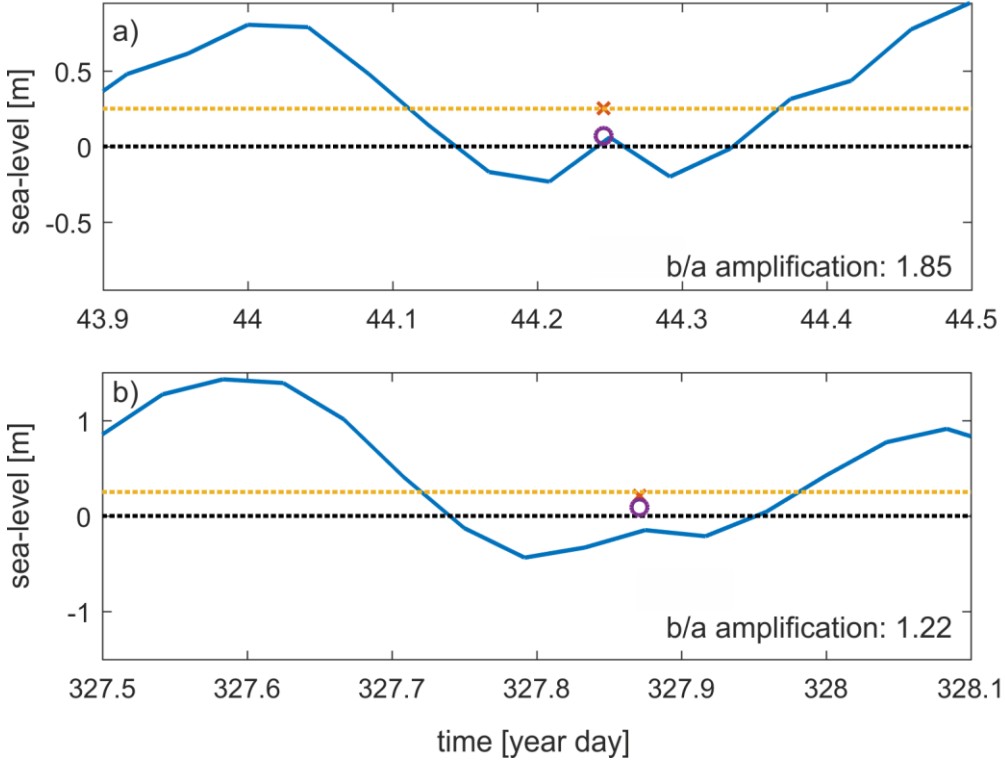

**Figure 5: Blow up of 13 hours of data from Providence for two different dates. The b/a ratio is plotted with an orange x; the orange dotted line marking he critical value form Doodson's criterion. The black circle and line mark the M6 amplitude at Providence and the 0 line. The text is the amplification of the b/a ratio between Newport and Providence based on a tidal fit to the period of data shown.**




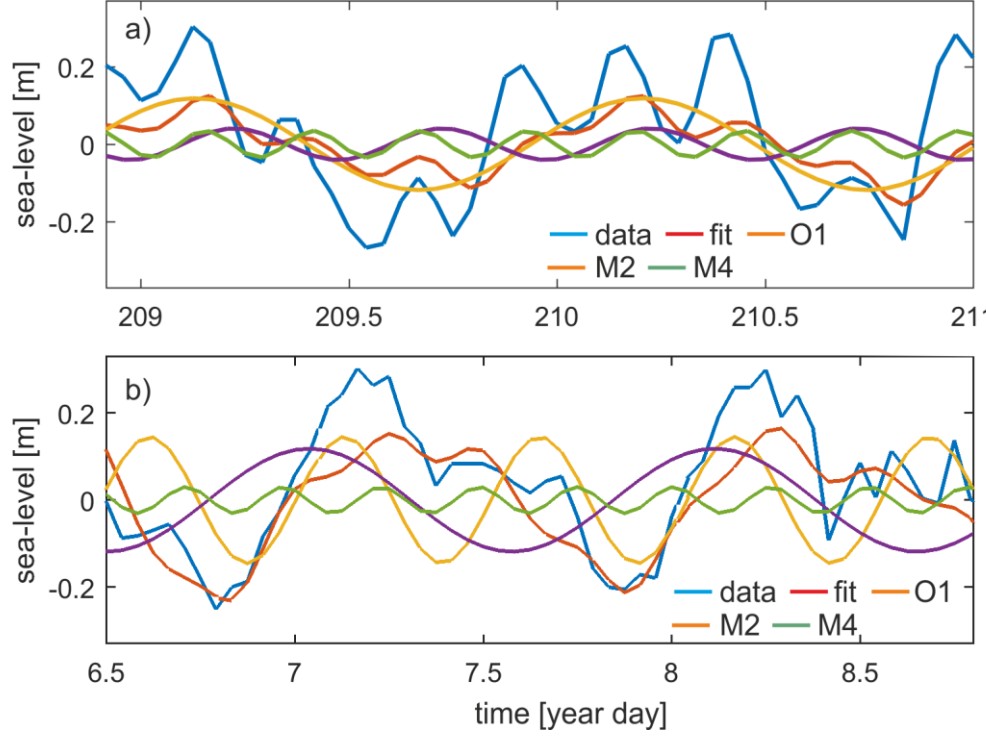

297

**Figure 6: Shown are two days of data from Rio Grande (panel a) and Imituba (panel b). Note the triple high in the Rio Grande series.**

298