# Peer review of "A mechanistic classification of double tides"

_Ocean Science, 2018_

## Referee Comment (RC1) · Anonymous Referee #1 · 28 Jul 2018

General comments

This is a well-structured manuscript, well-illustrated and referenced, though for a reviewer there are several irritating editorial blunders.

However, the authors overate the scientific interest in tidal double high and low waters, which are part of a wider context of higher harmonics distorting pure tidal cosine curves. A volume celebrating 100 Years of Liverpool Tidal Institute should hope for more significant contributions. This paper needs much more physical interpretation and scientific insight: we need more about the dynamics of Kelvin waves and Standing waves.

Classically Double Tides are found where an ingoing Kelvin wave and a weakened

outgoing reflected Kelvin wave interfere. Any classification must evolve from this basic well-understood hydrodynamics. Almost always the tidal ranges are small. Oddly, one of the best known and understood of these DT locations, Courtown on the eastern Irish coast, is not even mentioned in the paper, though very near to Bangor University!

In summary, If space is limited then it has a lower priority. But it could be published after substantial additional work if Editors have space available

Specific Comments

The paper has "mechanistic" in the title, which encapsulates the authors' approach. This is an exercise in number generation from a global tidal model, rather than development of a physical insight.

The elevation of a small piece of algebra in Doodson's Manual, to a "Doodson Criteria" is excessive. Doodson, surely the greatest contributor to the LTI -100 reputation, only showed an absolute minimum condition for DTs. Somehow the authors have set the "Criterion" up, so they can convieniently elaborate it by introducing phase relationships as a further refinement. . .more a complication than a new perspective.

Any discussion of DTs should look also at the dynamics of currents in the vicinity.

As the generation of shallow-water harmonics is a highly non-linear process, so too is their effect in distorting tidal curves. This is well covered in existing textbooks, despite the authors' assertion otherwise. And as a consequence of the non-linearity, DTs are much more evident at spring tides than at neaps. The paper should acknowledge this and deal with the temporal variations in DT occurrences (they do briefly and superficially for the 18.6 year nodal variations at the end of the paper) The other spring-neap influence on multiple HW/LW is the movement of amphidromes, as shown at Courtown (see above). The spring-neap effects on shallow water tides is basically caught by the MS4 harmonic constituent, which is not included in this paper.

Abstract L8 harmonics FIT to tidal curves. They do not "explain" in any physical sense.

L18 surely given the resolution of the model, these 8000 potential locations are generally in a much smaller set of local clusters.

L19 a curious use of "manual work"

L23 not per se especially scientifically interesting

L31 "transient coherent seiche"

L34 "Provides"

L37 and of course Courtown

L43 b/a

L53 discuss the dynamics behind this node/antinode configuration

This three classification categories are not robustly independent, so many cases may be a mixture of two or all three trigonometric criteria.

L63 "for the harmonic amplitude conditions"

L68 misuse of "still" . delete.

A more direct approach would be to examine a global tidal map for standing waves reflected in shallow water. (Remembering of course the spring-neap cycle)

L76 These are very small M2 tides, surely often hidden in non-tidal sea level changes. Classes 1 and 2 have a linked dynamic interpretation in terms of Kelvin waves, of which the Authors seem unaware.

L78 This thirs criterion misses the case where a tidal standing wave, say M4, has a node at the entrance, so zero M4 amplitude but strong M4 currents.

L85 MS4 (and maybe eg combinations of (M + S)6 etc) is essential here.

L01 Were these all independent dynamically? Could list.

L13 page 3 needs discussing in Kelvin Wave dynamics, relating to "the nature of the tide in the North Sea". It's the weaker reflected wave phenomena again.

Page 4 L53 does acknowledge the transient nature of DTs, but should consider these in relation to the S/N cycle. MS4 must be included.

L63 could be an amplitude node for M6…but not for currents. M4 , M6 resonances could lead to such patterns.

L68 suffixes missing

L70 Providence . "likely to be the " but this is a weak unsupported assertion.

L80 S/N much more important influence than Nodal effects.

L84 These are non-linear processes so the scaling is not linear. This argument needs refining or removing. Remember that the theoretical 3.7% nodal modulation is reduced to ca 2.2% in many places due to non-linear energy dissipation.

L89 Table 1

L 94 could refer to N**2 here and comment that n=2 still applies!

Page 6 L01 Omit this hand waving argument or clarify.."this may exaggerate" is very weak.

L11 A sweeping and erroneous generalisation, Although for illustration the M2/M4 example has merit in text books. Students do not need further complications, only maybe needing to be aware of the local nature of higher-harmonic generation. DTs occur in time not frequency space.

L17 delete "at the end of the day" jargon.

L22 on the climate change band wagon. Even the main tides are little affected by realistic sea level increases.

The Figures and Tables are ok, except that curves are clipped top and bottom in Figure

3.

---

## Referee Comment (RC2) · Anonymous Referee #2 · 6 Aug 2018

General Comments

This is a very poorly presented manuscript. It contains so many clumsy mistakes that I worry that there may be serious errors in the underlying calculations - errors that cannot be identified by a straightforward review process such as this. Some of the authors' arguments and discussions are so marred by inconsistencies and obvious errors that, in places, I got quite lost. An effective review of the scientific content is therefore not possible until most of these problems are resolved.

While the subject is of marginal interest, the authors do little to convince the reader that their findings have much use - the only statement I could find in this regard is on lines 223-234: "This has implications for mitigation purposes, because a prolonged high tide due to higher harmonics has the potential to increase flood risk due to storm surges".

[Figure]

The authors could do a lot better in "selling" their work.

The manuscript would require substantial revision and thorough checking prior to any re-review.

Specific Comments

Lines 31-33 ("The DT can be generated ..... low tide as well"): The logic of these two sentences isn't clear.

Lines 44-46: The criterion of Byrne et al. (2017) needs clarifying - there seem to be two semi-arbitrary constants (B and r) - I assume there must be a constraint on one of these constants - what is it?

Lines 44-45: The phrase "revised parameter" is curious, given that Equation (1) (Doodson's criterion) does not contain a parameter.

Line 55: I think "the semi-diurnal tide is reduced at neaps because M2 and S2 are about the same size" should be "the semi-diurnal tide is substantially reduced at neaps because M2 and S2 are about the same size" - by definition, "the semi-diurnal tide is reduced at neaps" by the interaction of M2 and S2, no matter what their relative size is.

Line 58: I don't understand the logic of "In the third class, M4 is amplified more than M2 inside an embayment and cant therefore generate a DT" - perhaps "cant" is meant to be "can"?

Lines 76-80: The criteria definiting these Classes seem to come from nowhere - they need to be explained and quantitatively justified.

Lines 103-104: I'm confused by "the locations of these gauges are marked in Figure 1b". Presumably this refers to the seven gauges referred to in Table 1. If the white crosses are the DT locations identified from TPXO, where are the tide-gauge locations marked?

Lines 113-116: This explanation is confusing. What does a "tidal minimum" (line 114)

mean? - presumably it means a minimum in M2 amplitude. Also, the location of Den Helder is most probably not at a "local minimum in M2 tidal amplitudes", because, by definition, the M2 amphidromes (one to the west of Den Helder and the other to the north-northeast) are points of zero M2 tidal amplitude (so the M2 amplitude at Den Helder must be larger than at either of the amphidromes - in fact, it looks as if it is near a local maximum, albeit a small one).

Line 116: I really don't see the point of Figure 2(c) - it certainly doesn't support the authors' case. While the blue curve (the observations) does indicate a weak double high tide (masked somewhat because the data is only hourly), the "reconstruction" (red curve), which includes M2 and M4, which supposedly give a double tide at this location, shows no such double tide - there is absolutely no explanation as to why this should be so.

Lines 124-125: I haven't a clue what is meant by "shows a flattened high tide during neaps, which M4 is able to modulate into the double low tide". For a start, M4 isn't "modulating" anything - it is just one harmonic component. Figure 3(a) doesn't show any double tide in either the observations (the blue curve) or the "reconstruction" (the red curve). There is a broadening of the observed high tide around day 3 but no actual double tide. Again, there is no explanation as to why the "reconstruction" shows absolutely no indication of a double tide.

Lines 126-128: the sentence "The mechanism is the same ..... the dominating diurnal tide" is unintelligible. And again, in Figure 3(b), there is no double tide in either the observations or "reconstruction" - the only deviation from a rather ordinary mixed semidiurnal/diurnal tidal curve are slight wiggles (presumable related to M4) in the observations just after days 275 and 276. There are just too much confusion for me to understand most of the paragraph in lines 124-129. If the authors want to be understood, they could at least indicate which of the points on the curves of Figure 3 that they consider to be double tides.

Line 136: instead of just saying that a b/a ratio of 0.18 "is not enough to produce a double low tide", it would be helpful if the authors referred back to Equation (1) which indicates that "b/a > 0.25" is the condition for the occurrence of double tides caused by an interaction of M2 and M4.

Lines 137-138: "This is because the annual fits ..... a double low in the Bay" is another inscrutable sentence.

Line 141: more detail is required here, concerning the "the fit on a 25-hour part of record". I assume (hope!) that, this was done only for one diurnal constituent, one semidiurnal constituent (presumably M2), M4, M6 and M8 - otherwise there would be a problem with splitting the constituent pairs K1 and O1, and M2 and S2.

Line 158: the word "supercritical" should be either defined or removed - in hydraulics, it has a quite different meaning to the meaning that is intended here - using the word in the present context only adds to the confusion.

Lines 158-159: what on Earth does "the phase is not right between the M2 low tide and the M4 high tide" mean?

Line 170: I don't understand how the fact that "the M6 amplitude in providence is $\sim$0.3ab" is consistent with M6 being "proportional to the product ab of M2 and M4 tides". All the authors have apparently done is to divide the M6 amplitude by the product of the amplitudes of M2 and M4, resulting in a value of about 0.3. This DOESN'T show that M6 is "proportional to the product ab of M2 and M4 tides" - all it does is show that, at a single location, the amplitudes M6/(M2*M4) are about 0.3.

Lines 183-185: an important point is missing in the sentence "the reason is quite simply ..... potential to generate DTs". The criterion for DTs is primarily based on ratios between the amplitudes of tidal components (e.g. M4/M2). However, the overtides, M4 and M6, are generated by nonlinear processes acting on the motions produced by pure lunar and solar forcing. Therefore, as M2 is reduced, so too are the ratios M4/M2 and

M6/M2, and the "potential to generate DTs" becomes smaller. Therefore, anything that reduces the amplitudes of major constituents tends to reduce the nonlinearities which produce the overtides which, in turn, tends to reduce the occurrence of DTs - which raises the question as to why, in this instance, the number of DTs isn't reduced.

Table 1: I don't know why the phases are given as "the phase ..... measured relative to the starting point of the time series". As such they are almost meaningless unless the date and time of each "starting point of the time series" is known; the phases given cannot be checked against existing estimates and the difference between the phases of different constituents only has any meaning when the constituents are harmonically related. The phases should, instead, be given in relation to the equilibrium tide (i.e. as the conventional harmonic constant, "g").

Figure 5 caption: it cannot be correct to say that "the b/a ratio is plotted with an orange x" because the vertical axis is sea level in metres - i.e. it can't be a ratio. "b" is the M4 amplitude and "a" is either the M2 amplitude or the semi-diurnal neap amplitude - so I can't understand how plotting "b", or "a", or "b/a", or some critical value of "b/a", on top of the observed sea level makes any sense.

Technical Corrections

Line 32: "higher harmonics" should be "higher harmonic".

Line 46: "For" should be "for".

Line 58: "cant" should be "cannot" or "can't" (but see earlier comment on Line 58).

Line 78: "ration" should be "ratio".

Line 99: "where" should be "were".

Line 103: "Table 2" should be "Table 1".

Line 107: "amphidromie" should be "amphidrome" or "amphidromic point".

Line 113: "to" (at right-hand end of line) should be "two".

Line 122: "at" should be "in".

Lines 146-147: the sentence "This further stress .... a double low tide" needs rewriting.

Line 170: "providence" should be "Providence".

Line 189: "Table 2" should be "Table 1".

Table 2 is not referred to in the text (except incorrectly when the authors mean "Table 1").

Figures showing maps: it is a pity that these are all bitmaps rather than vector graphics (e.g. Postscript or PDF), as the relatively poor resolution of the present figures does not allow much in the way of zooming in on specific features.

Caption to Figure 1: Presumably the white circles in (a), the white crosses in (b) and the red dots in (c) are the identified DT locations - if so, the caption should say so. Also, most of the "140 potential class 2 locations" seem to have disappeared from (b), presumably underneath the black triangles. The "the locations of these gauges ... marked in Figure 1b" (lines 103-104) also seem to be missing from both the figure and this caption.

Figure 2a and its caption, Table 1 and four places in the text (lines 36, 102, 109 and 111) : Should the location be called "Den Helder" or "den Helder"? - the authors need to be consistent.

Caption to Figure 2(a): this includes a reference to Figure 2(b) (the "right" panel). The captions to Figures 2(a) and 2(b) need to be rewritten to remove the confusion.

Figures 2(a) and 2(b): these maps are almost unreadable - for example, the M4 cotidal lines are very indistinct, especially in the region around Den Helder. Also, the scales for latitude and longitude severely distort features on the map (i.e. it is distinctly non-conformal) because one degree of longitude is mapped to a greater distance than

one degree of latitude, which is opposite to the way required for a conformal projection (such as a Mercator projection). It isn't very hard to make the shape of the land features roughly correct.

Figure 3: it is a pity that some of the curves are truncated at the maximum and minimum levels.

All figures of time series: some key colours are incorrect. For example, in Figures 2(c), 3, 4 and 6(b), the key shows O1 as orange but the O1 curve is in fact purple. In Figure 6(a), the key shows M2 as orange but the M2 curve is in fact purple.

All figures of time series: the vertical axis is marked "sea-level", which is an adjective - the noun, "sea level" (without a hyphen) should be used instead.

---

## Editor Comment (EC1) · P.L. Woodworth (Editor) · 25 Aug 2018

Detailed comments from the topic editor on 'A mechanistic classification of double tides' by Green et al. in Ocean Science Discussions.

I had independently made my own review of the paper as topic editor, which I enclose below. I hope that they complement those already made by R1 and R2.

This short paper uses a modern ocean tide model and a global tide gauge data set to determine where, and under what circumstances, that double high or low tides occur. It extends earlier work at Port Ellen published in this journal. I have a number of comments on the paper, although some of them are to do with the text and are easily fixed.

[Figure]

First, I am surprised that only 13 sites were identified globally. Pugh (1987), Pugh and Woodworth (2014) and Woodworth (J Geodesy 2017) mention a number, and I would have expected there to be over 13 on the south coast of England and adjacent European coasts alone. Also, another example, Reviewer 1 mentioned Courtown which has double highs now and again - see Figure 1.2 of Pugh and Woodworth (2014). The fact that we know of others which are not identified here presumably means that the criteria chosen for the classes are not optimum.

For example, lines 33-35 say:

This means that any location with a double high tide has a nearby location with a double low tide as well. [which is probably correct] Arguably, the best known double tide occurs in Southampton (UK), where the prolonged high water associated with the double low tide [this should read 'double high tide'] provided the port with a commercial advantage.

which is also correct. So it would be nice here to mention Weymouth for example as the double-low counterpart of S'ton, and to include it in the list (but possibly it does not fit into the criteria).

Second, I don't agree completely with the statements (e.g. line 19 and elsewhere) about 'manual work' always being needed. I have nothing against manual confirmation, but it is straightforward to write software to detect double tides. That software can work on, say, 1-minute predictions from the fits, even if the original data is hourly. Sometimes the double highs (lows) are marginal, when the central low (high) is just a mild inflection, but criteria can be written for selecting them - see for example the discussion of NOAA methods developed by Steacy Hicks in Section 5.3 of Woodworth (2017).

However, these things aside, one problem I had with the paper was the book-keeping:

(1) line 14 says 13 cases were found across the classes. It would be good to say clearly how many in each class. Even after reading the Results section and looking at figure 1 I was unclear where they were. If there are as few as 13 then it would be good to have a table listing them all.

(2) It seems to me that the sentence 'The search criteria' should come before 'Thirteen locations' as that is the logical order of the searching. And to make it clearer that one is now using tide gauge data instead of model, say something like 'Thirteen actual tide gauge locations were eventually ..'

(3) line 16 - says over 400 candidate locations from the model for classes 1 and 2, whereas line 96 says 219 and 140 which is not 'over 400'.

Line 32 - harmonic

- it would be good to give a reference to the S'ton example (Bowers et al., 2013 maybe?)

- Doodson proposed. Give a reference.

44-45 'The revised parameter is $B = br2/a$ ..' would be better expressed as something like 'The revised criterion is $b/a > 1/r2$ or $B > 1$, where $B = br2/a$ and r ...'

I have a little problem here in that the reader will have no idea what 'r' actually is, without reading Byrne et al. Maybe that is ok.

They –> Byrne et al. (2017)

when DTs occur (i.e. M6 is the source of the DT).

.. increase the ratio to meet the above criterion.

- class 1 - I am uneasy seeing S'ton mentioned here. The double high at S'ton is discussed by Pugh and Woodworth (2014) and Bowers et al. (2013) and by Doodson earlier, and it is clearly not a simple situation of M2 and M4. So I don't think this should refer to a textbook situation like this for S'ton.

- cant - this should be 'can' presumably.

- TPXO9 references should be given here when first mentioned.

The para has 'see below for details' twice which is a bit repetitive

I am not sure TPXO9 should be referred to 'altimetry data'. It is a model assimilation of the altimeter data. A pure altimeter map, which I guess is the input to the model, would look a bit different.

- I am not surprised resolving bays for the class 3 work was a problem when using a 1/6 deg model which is rather coarse. The FES2014 model is readily available which is 1/16 deg globally and is probably a generally better model as well.

- to avoid confusion it would be good to show the ratio as being M4/M2

This is really an extreme criterion choice with M4 much larger than M2, and M2 being selected as very small. There can't be many places like that, and this selection is possibly why you find so few candidates.

- should this read ratio between S2 and M2 (i.e. S2/M2 > 0.9)?

- ratio

I would reword '.. we obtain a lot of candidate points where this is fulfilled, but these occur (in the model) along straight coastlines instead (in reality) in bays or gulfs, due to the coarse model resolution. Consequently, this class requires further substantial ..

- GESLA would be better called GESLA-2

http://gesla.org should be https://www.gesla.org

- why were only 7 constituents considered? many double tides will be due to several constituents in the quarter and sixth diurnal bands.

- TPXO-data –> TPXO9 model.

- TG –> define acronym

- with the DTs found through visual inspection.

[But why? See my comment above]

For clarity, we only show data in the following from single 2-day periods during which DTs were found.

This is where I got really confused by the classes and numbers and what appears in the text and figure 1. Let's consider figure 1 first:

(i) all 3 panels repeat in showing M2 amplitude superimposed on which are black triangles for GESLA-2 sites. The triangles saturate coastal areas of interest where M2 is large (non-blue) such as the North Sea. So it looks just like a generally blue ocean with black edges.

I think I would have 2 separate panels showing the M2 amplitude in one alone, and the GESLA stations in the second.

(ii) then in (a), what are the white circles? The caption does not say. there does not seem to be 219 as you would expect from line 96. There are 2 arrows I can see - whereas the caption says arrows are for places discussed in the text: Denhelder, Rio Grande and Pari (lines 107-111) so the Pari arrow is missing.

then in (b), what are the dozen or so white crosses? The caption does not say. It has 3 arrows ok (Rio Grande, Thevenard and Victor Harbor I guess). I was expecting to see 140 dots for class 2 as mentioned on line 96.

then in (c), what are the red dots? The caption doess not say. One arrow for Nar Bay presumably.

I think this figure needs to be much better and the caption improved.

Back to the text:

96-105 - could this be reworded a bit to clean up the typos and make it clear how many stations in which class candidates from the model and then found in GESLA? So:

[Figure]

- .. something we did not take into account in the initial ..

- were then

- 'previously reported on'. what does this mean? what are the four? what reference? that is why it would be good to have a list in a table.

From these 13 [presumably], we opted ..

- Victor Harbor is usually spelt without the 'u'

Imtuba should be Imbituba.

put the countries after each as some people will not know where they are e.g. Den Helder (Netherlands).

.. are summarised ... locations of these seven [presumably] gauges ..

are marked in Figure 1b. But why? Are they all class 2?

- because an amphidrome

- between two

- below.

- in the data .. Harbor ..

- drop the hyphen

- define L. H is defined on page 1

- shown in Figure 5(a)

- is this 43.50 - 44.535 supposed to be the same as the 39-40.035 mentioned in Table 2 caption, and what looks like 43.9-44.5 in the figure? These bands need clarifying at least.

- mention again where this is (e.g., at Daya Bay, China, Song et al., 2016)

- visual inspection. see comments above.

- Figure 5(b)

- resonant period

- cycle influences

- actually, as odd as it sounds, S2 can have an apparent nodal variation in standard tidal analysis due to interactions with M2 etc. - see for example Figure 4 of Woodworth (2010, CSR) or a couple of papers by Amin.

- I don't think 'exciting' is a good word to use. Why and what does it excite? 'Interesting' maybe.

- Figure 7 should be 6(a)

define THT

the triples occur damped-down

- Imbituba also has triple highs (Figure 6(b)).

- M4 or M6 etc.

- again I am not sure visual inspection is so important, although doesn't do any harm. See above.

- and the south coast of England and Holland surely

- https://www.gesla.org

- tide gauge data contributors references - need doi adding for some.

- refers to

- this 'pha' value is not much use to anyone unless you also say when the starting point was, or give the Greenwich phase lag in the normal way.

- see comment for line 141

- it would be good to refer to the '25 hours' and 'annual neap' terms in the text where these things are discussed.

Say in the caption what 'Amplification' refers to.

Figure 1 - see above

Figure 2 and others: 'time [year day]' is not a good label as year does not appear in the scale. Perhaps 'time (day in year)' would be better.

caption (a,b):

The colours show M2 and M4 amplitudes from TPXO9 in (a) and (b) respectively, while the white (black) lines show the M2 and M4 Greenwich phase lags separated every 60 deg. The amphidromic .. arrows in (a).

but anyway I can't see any black lines, they are all white

.. O1, M2 and M4 curvevs respectively (see Table 1 ..

I think there has to be some statement that the colour scales in (a) and (b) saturate. For example M2 amplitude at Newhaven is over 2 m which is lot more than 0.2. There could also be an arrow on the colour scale at 0.2m.

Figure 2(c) and also later ones. As mentioned above, there is no reason why you have to plot the total fits and the individual terms as hourly values, even if the original data is hourly. You could for example plot them as 1-minute values which will make the double tides much easier to appreciate by inspection.

fig3 caption - (data from 2011) ... (data from 2004)

The fit here is a poor description of the data. Maybe that is what you are trying to show?

figure 4 - in (a) we have data for Providence in red, and in (b) the same data are shown in blue which is potentially confusing. I suggest the red and blue are switched in (a).

HA –> harmonic analysis. No point having acronyms if not necessary.

fig 5 caption - two different times in 20xx?

the critical value for fig 6 caption - Imbituba